# GULP: a prediction-based metric between representations

**Enric Boix-Adserà**
MIT
eboix@mit.edu

**Hannah Lawrence**
MIT
hanlaw@mit.edu

**George Stepaniants**
MIT
gstepan@mit.edu

**Philippe Rigollet**
MIT
rigollet@math.mit.edu

## Abstract

Comparing the representations learned by different neural networks has recently emerged as a key tool to understand various architectures and ultimately optimize them. In this work, we introduce GULP, a family of distance measures between representations that is explicitly motivated by downstream predictive tasks. By construction, GULP provides uniform control over the difference in prediction performance between two representations, with respect to regularized linear prediction tasks. Moreover, it satisfies several desirable structural properties, such as the triangle inequality and invariance under orthogonal transformations, and thus lends itself to data embedding and visualization. We extensively evaluate GULP relative to other methods, and demonstrate that it correctly differentiates between architecture families, converges over the course of training, and captures generalization performance on downstream linear tasks.

## 1 Introduction

The spectacular success of deep neural networks (DNN) witnessed over the past decade has been largely attributed to their ability to generate good representations of the data [BCV13] . But *what makes a representation good?* Answering this question is a necessary step towards a principled theory of DNN design. This fundamental question calls for a *metric over representations* as a basic primitive. Indeed, embedding representations into a metric space enables comparison, modifications and ultimately optimization of DNN architectures [LTQ+18]; see Figure 1.

In light of the practical impact of a meaningful metric over representations, this question has recently garnered significant attention, leading to a myriad of propositions such as CCA, CKA, and PROCRUSTES. Their relative pros and cons are currently the subject of a lively debate [DDS21, DHN+22] whose resolution calls for a theoretically grounded notion of metric.

**Our contributions.** In this work, we define a new family of metrics[1], called GULP[2], over the space of representations. Our construction rests on a functional notion of what makes two representations similar: namely, that two representations are similar if and only if they are equally useful as inputs to downstream, linear transfer learning tasks. This idea is partially inspired by feature-based transfer learning, in which simple models adapt pretrained representations, such as Inceptionv3 [SVI+16],

---

[1] More specifically, we define pseudo-metrics rather than metrics. However, these can be readily turned into a metric using metric identification. This amounts to allowing equivalence classes of representations.

[2] GULP is Uniform Linear Probing.

36th Conference on Neural Information Processing Systems (NeurIPS 2022).

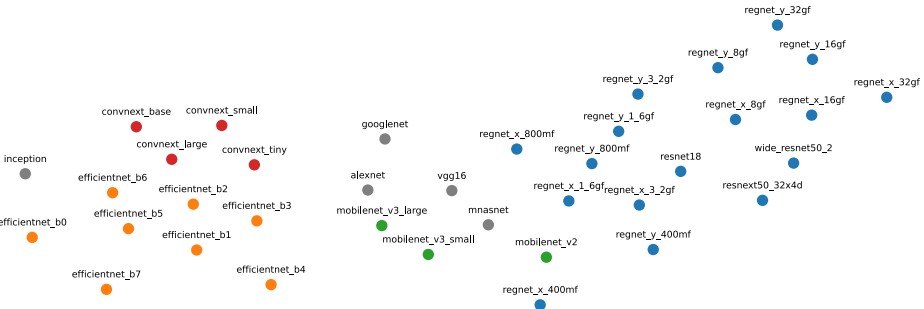

Figure 1: t-SNE embedding of various pretrained DNN representations of the ImageNet [KSH12] dataset with GULP distance ($\lambda = 10^{-2}$), colored by architecture type (gray denotes architectures that do not belong to a family). The embedding shows a good clustering of various architectures (ResNets, EfficientNets etc.), indicating that GULP captures intrinsic aspects of representations shared within an architecture family.

CLIP [RKH$^+$21], and ELMo [PNI$^+$18], for specific tasks [RASC14]; indeed, this is a key use for pretrained representations. Moreover, our application of *linear* transfer learning is reminiscent of *linear probes*, which were introduced by [AB17] as a tool to compare internal layers of a DNN in terms of prediction accuracy. Linear probes play a central role in the literature on hidden representations. They have been used not only to study the information captured by hidden representations [RBH20], but also to themselves define desiderata of distances between representations [DDS21]. However, previous applications of linear probing required hand-selecting the task on which prediction accuracy is measured, whereas our GULP distance provides a uniform bound over *all* norm-bounded tasks.

We establish various theoretical properties of the GULP pseudo-metric, including the triangle inequality (Thm 2), sample complexity (Thm 3), and vanishing cases. In particular, we show that akin to the PROCRUSTES pseudo-metric, GULP is invariant under orthogonal transformations (Thm 1) and vanishes precisely when the two representations are related by an orthogonal transformation (Thm 2).

In turn, we use GULP to produce low-dimensional embeddings of various DNNs that provide new insights on the relationship between various architectures (Figures 1, 5, and 6). Moreover, in Figure 7, we showcase a numerical experiment to demonstrate that the GULP distance between two independent networks decreases during training on the same dataset.

**Related work.** This contribution is part of a growing body of work that aims at providing tools to understand and quantify the metric space of representations [RGYSD17, MRB18, KNLH19, AB17, ALM17, CLR$^+$18, LC00, LV15, LYC$^+$15, LLL$^+$19, Mig19, STHH17, WHG$^+$18, DDS21, DHN$^+$22, CKMK22]. Several of these measures, such as SVCCA [RGYSD17] and PWCCA [MRB18], are based on a classical canonical correlation analysis (CCA) from multivariate analysis [And84]. More recently, centered kernel alignment CKA [CSTEK01, CMR12, KNLH19, DHN$^+$22] has emerged as a popular measure; see Section 2 for more details on these methods. The orthogonal procrustes metric (PROCRUSTES) is a classical tool of shape analysis [DM16] to compute the distance between labelled point clouds. Though not as conspicuous as CKA-based methods in the context of DNN representations, it was recently presented under a favorable light in [DDS21].

Various desirable properties of a similarity measure between representations have been put forward. These include structural properties such as invariance or equivariance [LC00, KNLH19], as well as sanity checks such as specificity against random initialization [DDS21], for example. Such desiderata can serve as diagnostics for existing similarity measures, but fall short of providing concrete design guidelines.

**Outline** The rest of the paper proceeds as follows. Section 2 lays out the derivation of GULP, as well as important theoretical properties: conditions under which it is zero, and limiting cases in terms of the regularization parameter $\lambda$, demonstrating that it interpolates between CCA and a version of CKA. Section 3 establishes concentration results for the finite-sample version, justifying its use in

practice. In Section 4 we validate GULP through extensive experiments[3]. Finally, we conclude in Section 5.

## 2 The GULP distance

As stated in the introduction, the goal of this paper is to develop a pseudo-metric over the space of representations of a given dataset. Unlike previous approaches, which work with finite datasets, we take a statistical perspective and formulate the population version of our problem. We defer statistical questions arising from finite sample size to Section 3.

Let $X \in \mathbb{R}^d$ be a random input with distribution $P_X$ and let $f : \mathbb{R}^d \to \mathbb{R}^k$ denote a *representation map*, such as a trained DNN. The random vector $f(X) \in \mathbb{R}^k$ is the *representation of $X$ by $f$*. We assume throughout that a representation map is centered and normalized, so that $\mathbb{E}[f(X)] = 0$ and $\mathbb{E}\|f(X)\|^2 = 1$. In particular, this normalization allows us to identify (unnormalized) representation maps $\phi, \psi$ that are related by $\psi(x) = a\phi(x) + b$, $P_X$-a.s. for $a \in \mathbb{R}$ and $b \in \mathbb{R}^d$, down to a single representation of $X$ (after normalizing), which is a well-known requirement for distances between representations [KNLH19, Sec. 2.3].

We are now in a position to define the GULP distance between representations; the terminology "distance" is justified in Theorem 2. To that end, let $\phi : \mathbb{R}^d \to \mathbb{R}^k$ and $\psi : \mathbb{R}^d \to \mathbb{R}^\ell$ be two representation maps, where $\ell$ may differ from $k$. Let $(X, Y) \in \mathbb{R}^d \times \mathbb{R}$ be a random pair and let $\eta(x) = \mathbb{E}[Y|X = x]$ denote the regression function of $Y$ onto $X$. Moreover, for any $\lambda > 0$, let $\beta_\lambda$ denote the population ridge regression solution given by

$$\beta_\lambda = \arg\min_\beta \mathbb{E}[(\beta^\top \phi(X) - Y)^2] + \lambda\|\beta\|^2$$

and similarly for $\gamma_\lambda$ with respect to $\psi(\cdot)$. Since we use squared error, these only depend the distribution of $Y$ through the regression function $\eta$.

**Definition 1.** *Fix $\lambda > 0$. The GULP distance between representations $\phi(X)$ and $\psi(X)$ is given by*

$$d_\lambda(\phi, \psi) := \sup_\eta \left( \mathbb{E}(\beta_\lambda^\top \phi(X) - \gamma_\lambda^\top \psi(X))^2 \right)^{\frac{1}{2}},$$

*where the supremum is taken over all regression functions $\eta$ such that $\|\eta\|_{L^2(P_X)} \leq 1$.*

The GULP distance measures the discrepancy between the prediction of an optimal ridge regression estimator based on $\phi$, and its counterpart based on $\psi$, uniformly over all regression tasks. While this notion of distance is intuitive and motivated by a clear regression task, it is unclear how to compute it *a priori*. The next proposition provides an equivalent formulation of GULP, which is amenable to accurate and efficient estimation; see Section 3. It is based on the following covariance matrices:

$$\Sigma_\phi = \mathrm{cov}(\phi(X)) = \mathbb{E}[\phi(X)\phi(X)^\top] \qquad \Sigma_\psi = \mathrm{cov}(\psi(X)) = \mathbb{E}[\psi(X)\psi(X)^\top] \qquad (1)$$

We implicitly used the centering assumption in the above definition, and the normalization condition implies that the covariance matrices have unit trace. Throughout, we assume these matrices are invertible, which is without loss of generality by projecting onto the image of the representation map. We also define the regularized inverses:

$$\Sigma_\phi^{-\lambda} := (\Sigma_\phi + \lambda I_k)^{-1}, \qquad \Sigma_\psi^{-\lambda} := (\Sigma_\psi + \lambda I_\ell)^{-1}$$

as well as the cross-covariance matrices $\Sigma_{\phi\psi}$ and $\Sigma_{\psi\phi}$ as follows:

$$\Sigma_{\phi\psi} = \mathbb{E}[\phi(X)\psi(X)^\top] = \Sigma_{\psi\phi}^\top. \qquad (2)$$

**Proposition 1.** *Fix $\lambda \geq 0$. The GULP distance between representations $\phi(X)$ and $\psi(X)$ satisfies*

$$\boxed{d_\lambda^2(\phi, \psi) = \mathrm{tr}(\Sigma_\phi^{-\lambda}\Sigma_\phi\Sigma_\phi^{-\lambda}\Sigma_\phi) + \mathrm{tr}(\Sigma_\psi^{-\lambda}\Sigma_\psi\Sigma_\psi^{-\lambda}\Sigma_\psi) - 2\,\mathrm{tr}(\Sigma_\phi^{-\lambda}\Sigma_{\phi\psi}\Sigma_\psi^{-\lambda}\Sigma_{\phi\psi}^\top)} \qquad (3)$$

*Proof.* See Appendix A.1. □

---

## 2.1 Structural properties

In this section, we show that GULP is invariant under orthogonal transformations and that it is a valid metric on the space of representations. We begin by establishing a third characterization of GULP that is useful for the purposes of this section; the proof can be found in Appendix A.1.

**Lemma 1.** *Fix $\lambda \geq 0$. The GULP distance $d_\lambda(\phi, \psi)$ between the representations $\phi(X)$ and $\psi(X)$ satisfies*

$$d_\lambda^2(\phi, \psi) = \mathbb{E}(\phi(X)^\top \Sigma_\phi^{-\lambda} \phi(X') - \psi(X)^\top \Sigma_\psi^{-\lambda} \psi(X'))^2,$$

*where $X'$ is an independent copy of $X$.*

We are now in a position to state our main structural results. We begin with a key invariance result.

**Theorem 1.** *Fix $\lambda \geq 0$. The GULP distance $d_\lambda(\phi, \psi)$ between the representations $\phi(X) \in \mathbb{R}^k$ and $\psi(X) \in \mathbb{R}^\ell$ is invariant under orthogonal transformations: for any orthogonal transformations $U : \mathbb{R}^k \to \mathbb{R}^k$ and $V : \mathbb{R}^\ell \to \mathbb{R}^\ell$, it holds*

$$d_\lambda(U \circ \phi, V \circ \psi) = d_\lambda(\phi, \psi)$$

*Proof.* We slightly abuse notation by identifying any orthogonal transformation $W$ to a matrix $W$ such that $W(x) = W \cdot x$. Note that for any representation map, we have $\Sigma_{W \circ f} = W \Sigma_f W^\top$ and

$$\Sigma_{W \circ f}^{-\lambda} = (W \Sigma_f W^\top + \lambda W W^\top)^{-1} = W(\Sigma_f^\top + \lambda I) W^\top = W \Sigma_f^{-\lambda} W^\top.$$

Hence, using Lemma 1, we get that

$$d_\lambda^2(U \circ \phi, V \circ \psi) = \mathbb{E}(\phi(X)^\top U^\top U \Sigma_\phi^{-\lambda} U^\top U \phi(X') - \psi(X)^\top V^\top V \Sigma_\psi^{-\lambda} V^\top V \psi(X'))^2$$
$$= \mathbb{E}(\phi(X)^\top \Sigma_\phi^{-\lambda} \phi(X') - \psi(X)^\top \Sigma_\psi^{-\lambda} \psi(X'))^2 = d_\lambda^2(\phi, \psi),$$

where we used the fact that $U^\top U = I_k$ and $V^\top V = I_\ell$. $\qquad\square$

Next, we show that GULP satisfies the axioms of a metric.

**Theorem 2.** *Fix $\lambda > 0$. The GULP distance $d_\lambda(\phi, \psi)$ satisfies the axioms of a pseudometric, namely for all representation maps $\phi, \psi, \varphi$, it holds*

$$d_\lambda(\phi, \phi) = 0, \qquad d_\lambda(\phi, \psi) = d_\lambda(\psi, \phi), \quad \text{and} \quad d_\lambda(\phi, \psi) \leq d_\lambda(\phi, \varphi) + d_\lambda(\varphi, \psi)$$

*Moreover, $d_\lambda(\phi, \psi) = 0$ if and only if $k = \ell$ and there exists an orthogonal transformation $U$ such that $\phi(X) = U\psi(X)$ a.s.*

*Proof.* Lemma 1 provides an isometric embedding of representations $f \mapsto f(X) \Sigma_f^{-\lambda} f(X')$ into the Hilbert space $L^2(P_X^{\otimes 2})$. It readily yields that $d_\lambda$ is a pseudometric. It remains to identify for which $\phi, \psi$ it holds that $d_\lambda(\phi, \psi) = 0$.

The "easy" direction follows from the invariance property of Theorem 1: if $\phi$ and $\psi$ satisfy $\phi(X) = U\psi(X)$ almost surely, then $d_\lambda(\phi, \psi) = d_\lambda(U\psi, \psi) = 0$. We sketch the proof of the other direction, and defer the full proof to Appendix A.2. Define $\tilde{\phi} = (\Sigma_\phi + \lambda I)^{-1/2} \phi$ and $\tilde{\psi} = (\Sigma_\psi + \lambda I)^{-1/2} \psi$. By Lemma 1, the condition that $d_\lambda(\phi, \psi) = 0$ is equivalent to $\tilde{\phi}(X)^\top \tilde{\phi}(X') = \tilde{\psi}(X)^\top \tilde{\psi}(X)$ almost surely over $X, X'$. So if $d_\lambda(\phi, \psi) = 0$, then we can leverage a classical fact that the Gram matrix of a set of vectors determines the vectors up to an isometry [HJ12], to prove that there is an orthogonal transformation $U \in \mathbb{R}^{k \times k}$ such that $\tilde{\phi}(X) = U\tilde{\psi}(X)$ almost surely over $X$. Finally, via analyzing a homogeneous Sylvester equation, this implies that $\phi(X) = U\psi(X)$ almost surely. $\qquad\square$

Note that when $\lambda = 0$, the conclusion of this theorem fails to hold: $d_0$ still satisfies the axioms of a pseudo-distance, but the cases for which $d_0(\phi, \psi) = 0$ are different. This point is illustrated in the next section where we establish that $d_0$ is the CCA distance commonly employed in the literature.

## 2.2 Comparison with CCA, ridge-CCA, CKA, and PROCRUSTES

Throughout this section, we assume that $k = \ell$ for simplicity.

**Ridge-CCA.** Our distance is most closely related to ridge-CCA, introduced by [Vin76] as a regularized version of Canonical Covariance Analysis (CCA) when the covariance matrices $\Sigma_\phi$ or $\Sigma_\psi$ are close to singular. More specifically, for any $\lambda \geq 0$, define the matrix $C_\lambda := \Sigma_\phi^{-\lambda} \Sigma_{\phi\psi} \Sigma_\psi^{-\lambda} \Sigma_{\psi\phi}$; the ridge-CCA similarity measure is defined as $\rho_{\lambda-\text{CCA}} = \text{tr}(C_\lambda)$. Hence, we readily see from Proposition 1 that GULP and ridge-CCA are describing the same geometry over representations. To see this, recall that Lemma 1 provides an isometric embedding $f \mapsto f(X)\Sigma_f^{-\lambda}f(X')$ of representation maps into $L^2(P_X^{\otimes 2})$. While GULP is the distance on this Hilbert space, ridge-CCA is the inner product.

Ridge-CCA was briefly considered in the seminal work [KNLH19] but discarded because of (i) its lack of interpretability and (ii) the absence of a rule to select $\lambda$. We argue that in fact, our prediction-driven derivation of GULP gives a clear and compelling interpretation of this geometry (as well as suggests several extensions; see Section 5). Moreover, we show that tunability of $\lambda$ is, in fact, a desirable feature that allows to represent the space of representations at various resolutions, giving various levels of information; for example, in Figure 6, higher $\lambda$ leads to a coarser clustering structure.

**CCA.** Due to the connection with ridge-CCA, our GULP distance is related to (unregularized) CCA when $\lambda = 0$. Specifically, defining $C := \Sigma_\phi^{-1} \Sigma_{\phi\psi} \Sigma_\psi^{-1} \Sigma_{\psi\phi}$, the mean-squared-CCA similarity measure is given by (see [Eat07, Def. 10.2]):

$$\rho_{\text{CCA}}(\phi, \psi) := \frac{\text{tr}(C)}{k} = 1 - \frac{1}{2k} \mathbb{E}\left[(\phi(X)^\top \Sigma_\phi^{-1} \phi(X') - \psi(X)^\top \Sigma_\psi^{-1} \psi(X'))^2\right],$$

where $X$ is an independent copy of $X'$; the last identity can be checked directly. From Lemma 1 it can be seen that our GULP distance $d_0(\phi, \psi)$ with $\lambda = 0$ is a linear transformation of $\rho_{\text{CCA}}$.

It can be checked that $\rho_{\text{CCA}}$ takes values in $[0, 1]$, which has led researchers to simply propose $1 - \rho_{\text{CCA}}$ as a dissimilarity measure. Interestingly, this choice turns out to produce a valid (squared) metric, i.e., a dissimilarity measure that satisfies the triangle inequality. Indeed, we get that

$$d_{\text{CCA}}^2(\phi, \psi) = 1 - \rho_{\text{CCA}}(\phi, \psi) = \frac{1}{2k} \mathbb{E}\left[(K(\tilde{\phi}(X), \tilde{\phi}(X')) - K(\tilde{\psi}(X), \tilde{\psi}(X')))^2\right]$$

where $K(u, v) = u^\top v$ is the linear kernel over $\mathbb{R}^d$ and $\tilde{\phi} := \Sigma_\phi^{-1/2}\phi$ (where $\tilde{\psi}$ and $\tilde{\phi}$ are the whitened versions of $\psi$ and $\phi$ respectively). These identities have two consequences: (i) we see from Lemma 1 that $d_{\text{CCA}}$ corresponds to the GULP distance with $\lambda = 0$ up to a scaling factor and (ii) $d_{\text{CCA}}$ is a valid pseudometric on the space of representations, since we just exhibited an isometry $T : \tilde{f} \mapsto K(\tilde{f}(X), \tilde{f}(X'))$ with $L^2(P_X^{\otimes 2})$. We show in Appendix A.2 that $d_{\text{CCA}}(\phi, \psi) = 0$ iff $\psi(X) = A\phi(X)$ a.s. for some matrix $A$. Note that the invariance of $\rho_{\text{CCA}}$ to linear transformations was previously known and criticized in [KNLH19] as arguably too strong.

**CKA.** In fact, thanks to the additional structure of the Hilbert space $L^2(P_X^{\otimes 2})$, the $d_{\text{CCA}}$ distance comes with an inner product

$$\langle T(\tilde{\phi}), T(\tilde{\psi})\rangle_{\text{CCA}} := \frac{1}{2k} \mathbb{E}[K(\tilde{\phi}(X), \tilde{\phi}(X'))K(\tilde{\psi}(X), \tilde{\psi}(X'))]$$

This observation allows us to connect CCA with CKA, another measure of similarity between distributions that is borrowed from classical literature on kernel methods [CSTEK01, CMR12] and that was recently made popular by [KNLH19]. Under our normalization assumptions, CKA is a measure of similarity given by

$$\rho_{\text{CKA}}(\phi, \psi) = \frac{\mathbb{E}[K(\phi(X), \phi(X'))K(\psi(X), \psi(X'))]}{\sqrt{\mathbb{E}[K(\phi(X), \phi(X'))^2]\,\mathbb{E}[K(\psi(X), \psi(X'))^2]}}$$

$$= \frac{\langle T(\phi), T(\psi)\rangle_{\text{CCA}}}{\|T(\phi)\|_{\text{CCA}}\|T(\psi)\|_{\text{CCA}}} = \cos\left(\measuredangle(T(\phi), T(\psi))\right),$$

where $\|T\|_{\text{CCA}}^2 = \langle T, T\rangle_{\text{CCA}}$ and $\measuredangle$ denotes the angle in the geometry induced by $\langle\cdot, \cdot\rangle_{\text{CCA}}$. In turn, $d_{\text{CKA}}^2$ is chosen as $d_{\text{CKA}}^2 = 1 - \rho_{\text{CKA}}$, which does not yield a pseudometric. This observation highlights two major differences between CCA and CKA: the first measures inner products and works with whitened representations, while the second measures angles and works with raw representations. As illustrated in the experimental section 4 as well as in [DDS21], this additional whitening step appears to be detrimental to the overall qualities of this distance measure.

The fact that GULP with $\lambda = 0$ recovers $d_{\text{CCA}}$ (i.e. $d_0^2 = 2k d_{\text{CCA}}^2$) is illustrated in Figure 2. As shown, although GULP has a roughly monotone relationship with CKA, they remain quite different.

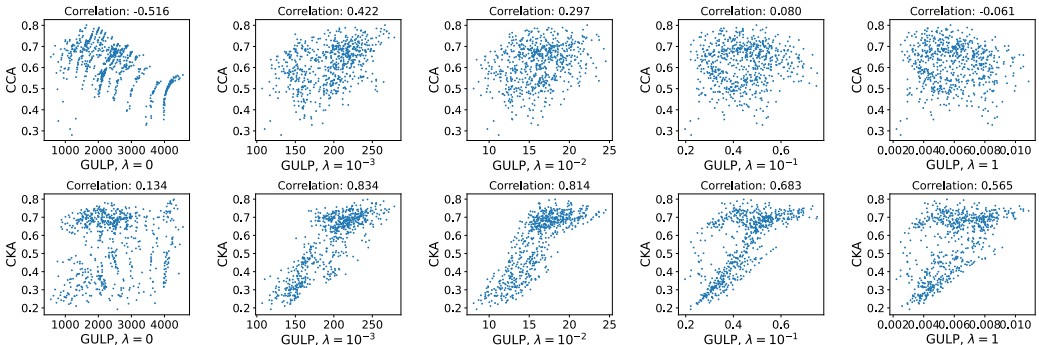

Figure 2: Empirical relationship between distances. Each point in the scatter-plot corresponds to a pair of ImageNet representations; the $x$-coordinate is the GULP distance, and the $y$-coordinate is the CCA or CKA distance. Although CCA and GULP for $\lambda = 0$ are related, their relationship is not linear since the representations' dimensionalities differ. Although CKA and GULP are related for large $\lambda$, their relationship is not linear due to the difference in normalization. Appendix B.2 contains more details and comparisons, including a surprisingly strong correlation between GULP and PROCRUSTES for some values of $\lambda$.

**PROCRUSTES.** The relationship between GULP and PROCRUSTES is not as clean as in the previous comparisons, but we include it for completeness. In the limit of infinite samples, the Procrustes distance as derived by [Sch66] is

$$d_{\text{Procrustes}} = \text{tr}(\Sigma_\phi) + \text{tr}(\Sigma_\psi) - 2\,\text{tr}\left((\Sigma_{\phi\psi}\Sigma_{\phi\psi}^\top)^{1/2}\right).$$

Our normalization implies $\text{tr}(\Sigma_\phi) = \text{tr}(\Sigma_\psi) = k$. However, the term $\text{tr}\left((\Sigma_{\phi\psi}\Sigma_{\phi\psi}^\top)^{1/2}\right)$ (which is equal to the nuclear norm $||\Sigma_{\phi\psi}||_*$) is not directly comparable to the preceding distances.

## 3 Plug-in estimation of GULP

In practice, the distribution $P_X$ of $X$ is unknown, so we cannot compute the population version of GULP exactly. Instead, we have access to a sample $X_1, \ldots, X_n \overset{\text{i.i.d.}}{\sim} P_X$. In all of the experiments of this paper, we approximate GULP with the following plug-in estimator:

$$\hat{d}_{\lambda,n}^2(\phi,\psi) = \text{tr}(\hat{\Sigma}_\phi^{-\lambda}\hat{\Sigma}_\phi\Sigma_\phi^{-\lambda}\hat{\Sigma}_\phi) + \text{tr}(\hat{\Sigma}_\psi^{-\lambda}\hat{\Sigma}_\psi\Sigma_\psi^{-\lambda}\hat{\Sigma}_\psi) - 2\,\text{tr}(\hat{\Sigma}_\phi^{-\lambda}\hat{\Sigma}_{\phi\psi}\hat{\Sigma}_\psi^{-\lambda}\hat{\Sigma}_{\phi\psi}^\top),$$

where

$$\hat{\Sigma}_\phi = \frac{1}{n}\sum_{i=1}^n \phi(X_i)\phi(X_i)^\top, \quad \hat{\Sigma}_\psi = \frac{1}{n}\sum_{i=1}^n \psi(X_i)\psi(X_i)^\top, \quad \text{and} \quad \hat{\Sigma}_{\phi\psi} = \frac{1}{n}\sum_{i=1}^n \phi(X_i)\psi(X_i)^\top$$

are the empirical covariance and cross-covariance matrices, and

$$\hat{\Sigma}_\phi^{-\lambda} = (\hat{\Sigma}_\phi + \lambda I)^{-1}, \quad \text{and} \quad \hat{\Sigma}_\psi^{-\lambda} = (\hat{\Sigma}_\psi + \lambda I)^{-1}$$

are the empirical inverse regularized covariance matrices. To justify our use of the plug-in estimator, we prove concentration around the population GULP distance as $n$ goes to infinity.

**Theorem 3.** *Assume that $\|\phi(X)\|^2, \|\psi(X)\|^2 \le 1$ almost surely. Then, for any $\lambda \in (0,1)$, $\delta > 0$, with probability at least $1 - \delta$ the plug-in estimator $\hat{d}_{\lambda,n}^2$ satisfies*

$$\left|\hat{d}_{\lambda,n}^2(\phi,\psi) - d_\lambda^2(\phi,\psi)\right| \lesssim \frac{1}{\lambda^3}\sqrt{\frac{\log((k+l)/\delta)}{n}}.$$

We defer the proof of this theorem to Appendix A.3. At a high-level, we first show that the inverse regularized covariance matrices, $(\Sigma_\phi + \lambda I)^{-1}$ and $(\Sigma_\psi + \lambda I)^{-1}$, are well-approximated in operator norm, so the expectation of the plug-in estimator is close to the population distance. We then apply

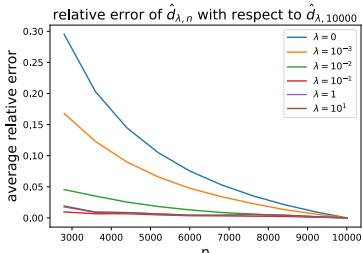

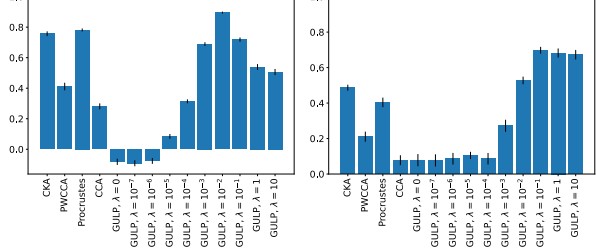

Figure 3: Convergence of plug-in estimator as $n \to \infty$. We plot relative error $|\hat{d}_{\lambda,n}^2 - d_{\lambda,10000}^2|/d_{\lambda,10000}^2$ averaged over pairs of ImageNet DNNs.

Figure 4: GULP captures generalization of linear predictors. We plot Spearman's $\rho$ between the differences in predictions by $\lambda$-regularized linear regression, and the different distances. Results are averaged over 10 trials.

McDiarmid's inequality to show that the plug-in estimator concentrates around its expectation. Note that the boundedness conditions on the representations are here to simplify technical arguments by appealing simply to McDiarmid's inequality; these can be presumably be relaxed to weaker tail conditions at the cost of more involved arguments.

Figure 3 supports our theoretical result by showing convergence on pairs of networks on the ImageNet dataset. See Appendix B.3 for more details.

## 4 Experiments

We evaluate our distance in a variety of empirical settings, comparing to CCA, CKA, the classical PROCRUSTES method from shape analysis, and a variant of CCA known as projection-weighted CCA (PWCCA); see [DDS21, Sec. 2] for definitions.

### 4.1 GULP captures generalization performance by linear predictors

The GULP distance is motivated by how differently linear predictors using the representations $\phi$ and $\psi$ generalize. In this section, we demonstrate that GULP indeed captures downstream generalization performance by linear predictors. We consider the representation maps $\phi_1, \ldots, \phi_m$ given by $m = 37$ pretrained image classification architectures on the ImageNet dataset $P_X$ (see Appendix B.5). For each pair of representations, we estimate the CKA, CCA, PWCCA, and GULP distances, using the plug-in estimators on 10,000 images, sufficient to guarantee good convergence (see Figure 3).

We then draw $n = 5,000$ images from the dataset $X_1, \ldots, X_n \sim P_X$, and assign a random label $Y_k \sim \mathcal{N}(0,1)$ to each one. For each representation $i \in [m]$, we fit a $\lambda$-regularized least-squares linear regression to the training data $\{(X_k, Y_k)\}_{k \in [n]}$, which gives a coefficient vector $\beta_{\lambda,i}$. Finally, for each $1 \leq i < j \leq m$, we estimate the distance $\tau_{ij} = \mathbb{E}_{X \sim P_X}[(\beta_{\lambda,i}^\top \phi_i(X) - \beta_{\lambda,j}^\top \phi_j(X))^2]$ between the predictions with representations $\phi_i$ and $\phi_j$, by taking the empirical average over 3000 samples in a test set. In Figure 4, we plot Spearman's $\rho$ rank correlation between $\tau$ and each of the distances GULP, CKA, CCA, PWCCA, viewed as vectors with $\binom{m}{2}$ entries, one for each pair of networks. Notice that for each $\lambda$, the distance that attains the best correlation is the GULP distance with that $\lambda$. This indicates that while GULP is a measure of distance that holds uniformly over prediction tasks, it retains its meaning in the context of a single prediction task.

### 4.2 GULP distances cluster together networks with similar architectures

We are interested in how GULP can be used to compare networks of different architectures trained on the same task. We begin by comparing fully-connected ReLU networks of widths ranging from 100 to 1,000 and depths ranging from 1 to 10, trained on the MNIST handwritten digit database. Every architecture is retrained four times from different initializations (see Appendix B.1). We input all MNIST training set images into each network, save their representations at the final hidden layer, and compute CKA, PROCRUSTES, and GULP distances between all pairs of representations.

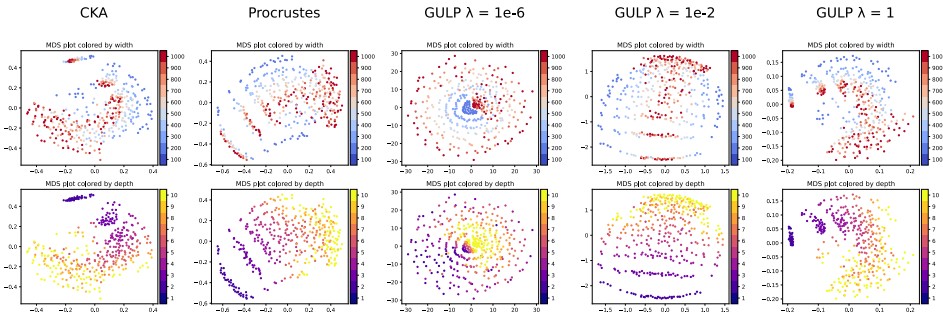

Figure 5: Two dimensional MDS embedding plots of fully-connected ReLU networks colored by architecture width (top) and depth (bottom). Networks are fully-trained on MNIST and penultimate layer representations are constructed from 60,000 input train images.

Figure 5 shows Multi-Dimensional Scaling (MDS) embeddings of the distances between all MNIST networks, color coded by width and depth. For GULP with $\lambda = 10^{-6}$, the networks are largely organized according to rank of the feature matrix: networks of large width and small depth, ones whose representations have the largest rank, are the most different, as evidenced by the halo of points in the MDS plots. This outcome confirms that CCA simply measures rank [KNLH19]. However, for PROCRUSTES and GULP with $\lambda = 10^{-2}$, networks become clustered by their depth, as evidenced by the striations in the MDS embeddings (plot colored by depth). Furthermore, networks of the same depth look most similar at large widths, as shown by the red centerline in the MDS embedding (plot colored by width), implying that as width increases networks converge to a shared limiting representation. Finally, GULP with $\lambda = 1$ closely resembles CKA and roughly organizes networks by depth. A takeaway is that GULP with $\lambda = 1$ resembles PROCRUSTES and CKA, and captures intrinsic characteristics such as width and depth.

Next, we show how distances between penultimate layer representations allow us to cluster pretrained networks with more complex architectures and, in turn, draw comparisons between them. To that end, we study 37 state-of-the-art models on the ImageNet Object Localization Challenge, of which the four major groups are ResNets, EfficientNets, ConvNeXts, and MobileNets (see Appendix B.1).

We compute the baseline distances between every pair of representations using 10,000 training images, and visualize them using a two-dimensional t-SNE embedding in Figure 6. Below each embedding plot we show the dendogram resulting from a hierarchical clustering of the networks based on their distances. As seen from the embeddings, when $\lambda$ increases, the GULP distance separates the ResNet architectures (blue) from the EfficientNet and ConvNeXt convolutional networks (orange and red). Compared to other distances, GULP with large $\lambda$ is able to more compactly cluster ResNets and convolutional networks separately. In Appendix B.5 we further quantify the compactness of clusterings under each distance metric by computing the standard deviation of distances within each cluster.

### 4.3 Network representations converge in GULP distance during training

So far, we have used GULP to compare static networks taken as a blackbox representation maps. Now we use GULP to examine how representation maps evolve over the course of training. To that end, we independently train 16 Resnet18 architectures on the CIFAR10 dataset [KH$^+$09] for 50 epochs. Figure 7 tracks the distance (averaged over all network pairs) at each epoch.

As shown, other distances change very little or even briefly increase over the course of training. For GULP with small $\lambda$, the previous sections have demonstrated that our distance captures fine-grained differences between representations; here too, it accentuates differences in representations mid-training (visible around epoch 25). However, as $\lambda$ increases, the GULP distance differentiates less between representations, and smoothly decreases over the course of training, thus indicating that it captures intrinsic properties of the representations rather than artifacts due to random seeds.

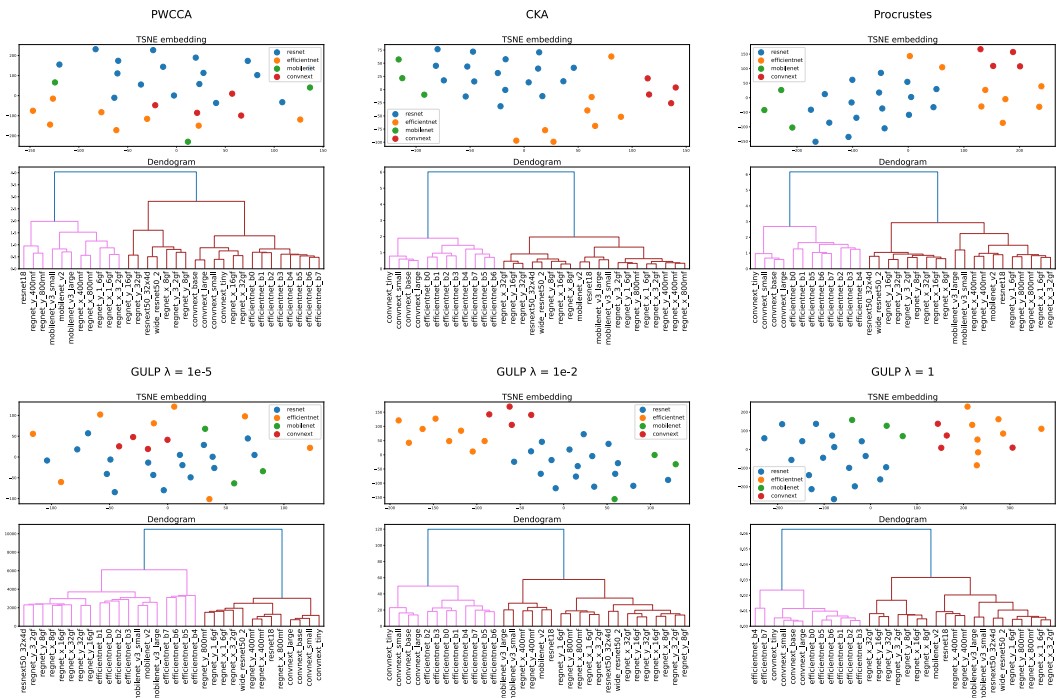

Figure 6: Embeddings of PWCCA, CKA, PROCRUSTES, and GULP distances between the last hidden layer representations of 36 pretrained ImageNet models (top) along with their hierarchical clusterings (bottom). All distance metrics separate ResNet architectures (brown dendogram leaves) from the rest of the ConvNeXt and EfficientNet architectures (pink dendogram leaves). GULP at $\lambda = 1$ is the most effective distance at separating ResNets from the remaining architectures.

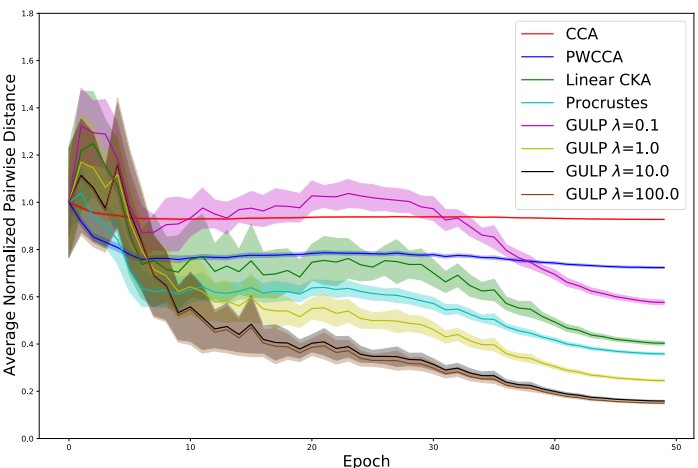

Figure 7: Empirical distances between representations of 16 independently trained ResNet18 architectures during training, computed using 3,000 samples and averaged over all $\binom{16}{2}$ pairs. Distances are scaled by their average value at iteration 0.

### 4.4 Sensitivity versus specificity of GULP

In Appendix B.9, we reproduce the experiments of [DDS21]. Our distance compares favorably to baselines and correlates with measures of a DNN's functional behavior. It achieves the specificity of CCA and PWCCA to random initializations, and improves the sensitivity of CKA to out-of-distribution performance.

## 5  Conclusion

In this paper, we have defined a family of distances for comparing learned representations in terms of their worst-case performance gap over all $\lambda$-regularized regression tasks. We proved convergence of the finite-sample estimator of this distance, quantified its relationship to existing notions such as CCA, ridge-CCA, and CKA, and demonstrated promising performance in a variety of empirical settings, including the ability to distinguish between network architectures and to capture performance differences on regression tasks.

Further studying extensions beyond linear transfer learning could provide a rich direction for future work. In fact, preliminary experiments reported in Appendix B.10 indicate that, compared to section 4.1, GULP fails to predict generalization performance when the downstream task shifts from linear to logistic prediction. This suggests extending GULP to a uniform bound over other downstream predictive tasks, such as logistic regression, multi-class classification, or kernel ridge regression. Although GULP under kernel ridge regression has a closed form using the kernel trick[4], GULP for logistic regression does not have a closed form. This brings additional computational questions of interest that are beyond the scope of this work. Finally, it could be interesting to consider the application of GULP to knowledge distillation, or alternatively to consider adding a ridge regularization term to probing methods (inspired by GULP).

## Acknowledgments and Disclosure of Funding

EB is supported by an Apple AI/ML Fellowship, and the National Science Foundation Graduate Research Fellowship under Grant No. 1745302. HL is supported by the Fannie and John Hertz Foundation and the National Science Foundation Graduate Research Fellowship under Grant No. 1745302. GS is supported by the National Science Foundation Graduate Research Fellowship under Grant No. 1745302. PR supported by NSF awards IIS-1838071, DMS- 2022448, and CCF-2106377.

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
