# OpenReview forum: "GULP: a prediction-based metric between representations"
_NeurIPS.cc/2022/Conference — NeurIPS 2022 Accept_

### Official Review · Reviewer_zDqK · 2022-06-24

**Rating:** 6
**Confidence:** 3
**Soundness:** 3 good
**Presentation:** 2 fair
**Contribution:** 2 fair

**Summary:**

This paper introduces GULP (a pseudometric) for comparing the representations learned by two different neural networks. The core idea is to use a downstream task on the learned representations to derive the distance function. The performance on downstream tasks generally determines the usefulness of representation. So, I think it is a good idea to rely on a downstream task when comparing the representation space of two different networks.

Authors use ridge regression as a downstream task. Each sample is assigned a random target label drawn from the Normal distribution, and a linear ridge regression model is trained. The final distance is a difference in the regression estimate of two representations under all regression functions.

The authors present both theoretical and empirical results. The theoretical results derive the closed-form expression of distance and prove its structural properties. Authors use sample estimates for GULP and theoretically justify that the estimator is concentrated around its expectation. The empirical results show the benefits of GULP, such as it captures generalisation over linear predictors, can be used for comparing architectures, the evolution of distance over training, and the sensitivity analysis of networks.

While the work is interesting, I think the critical element missing is the justification behind the choice of distance. The distance is closely related to ridge-CCA. I find it hard to understand the introduced distance's essential theoretical benefits over other alternatives.

I am more concerned about the representation used for computing the distance. After the final layer of neural networks, representation retains class-specific information and throws away the rest. That is to say, in BCE loss, neural networks maximise MI(Z; Y), where Z is the representation and Y is the task. I do not see the usefulness in comparing two networks using a distance on their final layer representation for any different downstream task such as ridge regression. Perhaps comparing the representations learned by self-supervised/unsupervised methods will be more meaningful.
Overall I think it is a good paper with sound theoretical results. On the practical side, there is a scope to improve the paper. Can GULP be extended for different downstream tasks? Also, some justification for using ridge regression as a downstream task. I give a score of 6. If the authors respond sufficiently to my comments, I will be happy to adjust my score.

***Post Rebuttal***

I appreciate the authors for their effort in replying to my comments. I find it as a good paper that could be more attractive with the right set of empirical results. I prefer to stay with my original assessment.


**Questions:**

In section 2, line 71, the representation map 'f' is centred to 0 mean. Next, the maps phi and xsi are introduced. It is unclear whether phi and xsi are two choices for 'f' also centred to 0 mean. If yes, then I think its wrong to say ‘xsi(x) = a phi(x) + b’ as this would imply E[xsi(x)] = b whereas the normalisation would mean E[xsi(x)] = 0.

Can GULP be used for comparing the representation of networks trained on different datasets? It can be an interesting empirical result.

**Limitations:**

The authors provide some discussion on the limitations of their work. The main downside, in my view, is that GULP relies on the ridge regression task. If the network is trained with cross-entropy loss, it is not immediately obvious what are the benefits of evaluating it under a ridge regression task. Furthermore, downstream applications have other choices such as classification, clustering, etc. The authors mention that in conclusion, but it appears to be the main issue as much of the downstream tasks are not regression. They also say GULP has a closed-form solution under kernel ridge regression. It would be better if the authors showed empirical results with kernels.

The empirical evaluations use final layer representation would be interesting to investigate applications of GULP for comparing intermediate representations as well as its suitability for comparing networks trained on different datasets.

**Strengths And Weaknesses:**

Strengths:

Theoretically sound pseudometric for comparing representations of two architectures. The paper is well written, and the proofs are easy to follow.
Empirically demonstrate the use of GULP on applications such as grouping similar architectures, comparing the training dynamics, and the sensitivity analysis of architectures.

Weaknesses:

The analysis in section 4.2 relies on MNIST data. Given the complexity of features in MNIST digits, I wonder whether the information content of the features learned by various architectures differs significantly. A little more challenging data, such as cifar10, 100, etc., would be more realistic to consider. Authors later analyse architectures trained on the imagenet. However, I still think MNIST is not a good choice, even as a toy problem for comparing representation.

The empirical results can be better established. For instance, one can use GULP to compare representations of different layers of the same network to see how downstream performance improves with the depth or width of the network. Can potentially investigate its application to tasks such as knowledge distillation.

I don't understand the motivation and practicality of using GULP for network convergence. In section 4.3, 16 resnet18 models are trained, and their distance is reported under GULP and other distances. It is not apparent what the authors mean by distances capture fine-grained differences. The same model should learn similar features under different initialisation for the same task. One could report mean and std in the loss for this purpose. What is an added gain to looking at "fine-grained differences in features". After all, the distance comes at an additional computational cost.

Minor Comment: The choice of notation could have been better. It is easier if random variables are bold, likewise use a standard choice for scalar, vector, matrices, etc. The x-axis labels in dendrogram plots are too small, making them hard to interpret. It would be nice to plot these in higher resolution and move them to the appendix.

---

> ### Author Response · Authors · 2022-08-02
> **Response to Reviewer zDqK (1/2)**
>
> We thank the reviewer for their feedback, and respond to each point individually below.
>
> **Clarification on the ridge regression labels**
>
> Though not an explicit question, for the sake of correctness, we first wish to address the reviewer’s summary that “each sample is assigned a random target label drawn from the normal distribution”: the definition of GULP is actually worst-case over all norm-bounded target (label) functions $\eta$. The random target label is simply a heuristic we use in the experiments in Section 4.1, where it serves as a single downstream task. In a way, the experiment of Section 4.1 compares our definition, which is worst-case over downstream tasks, to an “average-case,” random downstream task.
>
> **Relationship to ridge-CCA**
>
> As we discuss in lines 130-135, GULP turns out to indeed describe the same geometry over representations as ridge CCA. However, we derive GULP in a completely novel way, inspired by the transfer learning applications of deep representations. This provides a natural interpretation of the regularization parameter $\lambda$, which we now view as a useful feature rather than a hindrance (by showing, in a variety of experiments, how $\lambda$ enables the user to tailor the distance measure’s resolution to the task at hand). The transfer learning interpretation of this geometry between representations also inspires a bevy of extensions, depending on the downstream task; we give a closed form for kernel ridge regression in this work, but extensions to logistic regression or shallow networks provide an interesting avenue for future work.
>
> **Meaningfulness of representation**
>
> The reviewer states, “I do not see the usefulness in comparing two networks using a distance on their final layer representation for any different downstream task such as ridge regression.” First we would like to clarify that in most of our experiments, the GULP distance is computed between representations at the penultimate (last hidden) layer, not the output layer of the network. In general, GULP is a method for comparing any representations, which do not necessarily need to come from the last hidden layer representation of a network. Indeed, we also include experiments applying GULP to intermediate network layers, such as in Appendix B.7. The key intuition behind GULP is simply that the value of data representations often lies in their distillation of important features about the data, in the sense that a relatively simple classifier trained directly on those representations suffices for many transfer learning tasks of interest. This motivates the use of simple ridge regression as a downstream task.
>
> **Extension for different downstream tasks**
>
> The reviewer asks, “Can GULP be extended for different downstream tasks”? The first paragraph of this response is relevant in answering this question. In particular, GULP is not restricted to a single downstream task, but in fact maximizes over all bounded-norm labeling distributions (which we call $\eta$ in the paper). Therefore, the GULP distance provably bounds the difference in ridge regression (transfer learning) performance between the two representations of interest over a very expressive family of downstream tasks.
>
> **Use of MNIST dataset**
>
> We agree that MNIST is far from capturing the complexity of more modern datasets. However, its moderate size allowed us to illustrate effects on training. Indeed, training many large architectures on large datasets requires much more compute power than what was employed in this work. We reiterate that this work is not about improving prediction performance (for which the mobilization of large computational power is better justified), but rather providing tools for researchers who do. To further validate the use of MNIST in many of our experiments, we replicated our original Figure 1 -- showing an embedding of many different architectures’ representations on ImageNet data, based on their GULP distances -- using the MNIST dataset instead in (new) Figure 18. As shown, using the simpler MNIST dataset still captures the key behavior we observed in Figure 1; namely, clustering representations by the architecture family which produced them. Moreover, several of our figures use CIFAR or ImageNet representations, including the newly added Figures 13 (CIFAR) and 14 (ImageNet).

---

> > ### Author Response · Authors · 2022-08-02
> > **Response to Reviewer zDqK (2/2)**
> >
> > **Experiment on the impact of depth/width on downstream performance**
> >
> > To first clarify, any distance between representations cannot predict whether downstream performance is “good” or “bad” in an absolute sense -- we can only say whether the performance of two representations should be similar, or not, on a downstream task. The experiment of comparing representations at different layers of the same network, while varying depth and width, is carried out in Ding et al. 2021 with competitors to GULP. One would expect similar results as predicted by the experiments (comparing two different networks) on Figure 11. Indeed, we have added a new Figure 21 comparing the intermediate representations of BERT networks on the MultiNLI dataset, as a function of network depth. We find that for each of the 10 BERT networks, the GULP distance arranges their hidden layers linearly in order from their input layer to their output layer and tuning the parameter $\lambda$ allows us to make distinctions between earlier and later layers of the network architectures.
> >
> > With that said, our empirical evaluations focus on computing distances based on the embedding of the last hidden layer, because it is common in transfer learning to run linear regression on a pretrained representation of data, and the penultimate layer representation is often used for this. This paradigm was one of the motivations of the definition of our distance.
> >
> > **Applications to knowledge distillation**
> >
> > This is a good suggestion, thank you. We will add a sentence on future application to knowledge distillation to the conclusion of a camera-ready version.
> >
> > **GULP for network convergence**
> >
> > Figure 7, which tracks the average distance between last hidden layer representations of several independently initialized CIFAR ResNets, is an exploratory experiment, rather than one motivated by practice. In particular, we view these results as a sanity check: the representations are much closer together by the end of training, than at the start. Moreover, smaller values for $\lambda$ capture small changes in the representations over the course of training (seen as bumps), which may be small fluctuations due to the random initialization, whereas larger values for $\lambda$ smoothly decay over the course of training. One could of course report the standard deviation of the loss as another useful metric, but this would capture something fundamentally different: namely, the loss of the ensemble of representations on the task at hand (in this case, CIFAR classification), rather than the difference in ridge regression performance across all downstream tasks (in some norm-bounded ball), which is what GULP captures. For instance, different representations may behave very differently on some task, which is distinct from the training task (CIFAR), but equally well on CIFAR. As a distance directly inspired by transfer learning, GULP captures these fine-grained differences. Finally, although GULP does come at an additional computational cost, it is minimal and scales like $O(N*k^2 + k^3)$, where $N$ is the number of samples used for the plug-in estimator in Section 3 and $k$ is the dimension of the two representations. (Of course, the representations need not have equal dimension, but we assume this for a simple runtime complexity calculation.) CCA and its variants also share the same computational cost as GULP.
> >
> >
> > **Centering of the representation (Question 1)**
> >
> > Specifically in Section 2 line 71, $\psi$ and $\phi$ are two unnormalized representation maps; we mean to say that if they are related by an affine transformation when normalized, they will become the same representation map after normalization. After establishing the normalization (i.e. from that point on), we assume $\psi$ and $\phi$ are normalized. We have clarified this in the revision.
> >
> > **GULP for comparing representations from different datasets (Question 2)**
> >
> > Yes, GULP can certainly apply to comparing the representations of networks trained on different datasets. This is an interesting question for future research in this direction.
> >
> > **Extensions to other downstream tasks than ridge regression**
> >
> > Indeed, in the Appendix A.4, we show that there is a closed-form solution for our distance under kernel ridge regression, and moreover that it is efficiently computable via the kernel trick. We agree with the Reviewer in viewing the extension of our theory and experiments to other prediction tasks, including kernels and logistic regression, as an exciting next step. However, to better convey our main point of prediction-driven distance measures, we have chosen to focus on arguably the simplest collection of predictive tasks, namely ridge regression. Undoubtedly, interesting phenomena will arise for more complex tasks in future work.

---

> > > ### Comment · Reviewer_zDqK · 2022-08-05
> > > **Response to the comments of authors**
> > >
> > > I express my thanks to the authors for replying to my comments. After reviewing the response, I think I will stay with my original assessment.
> > >
> > > I am mainly worried about its widespread impact and outreach to the community. The method is closely related to ridge-CCA. Although derivation takes inspiration from transfer learning, the end interpretation is similar with some perspective on regularisation parameter \lambda. I don’t see any substantial gain in using GULP over ridge-CCA on two feature matrices. I find this as a potential limitation from the novelty side. I think a more suitable space of experiments (such as comparing representation across datasets, knowledge distillation, other tasks using the kernel, etc.) could have established empirical benefits.
> > >
> > > Sorry, I meant penultimate layer when I mentioned “final layer”. I understand the motivation of ridge regression on the penultimate layer. My point was the penultimate layer representation of the network trained with target labels doesn’t benefit much in transfer. On the other hand, representation learned by self-supervised methods is more commonly suited for transfer.
> > >
> > > For the same architecture, I still find it hard to grasp the benefit of GULP in monitoring network convergence. I don’t see what extra information the ridge regression performance provides here. Perhaps an example of when a different representation of the same network can behave differently on the same task will be helpful.
> > >
> > > Overall I think it’s a good paper that could have been more attractive with suitable experiments.

---

### Official Review · Reviewer_d1FQ · 2022-07-09

**Rating:** 7
**Confidence:** 4
**Soundness:** 4 excellent
**Presentation:** 4 excellent
**Contribution:** 3 good

**Summary:**

Main Points:
- Provide a distance metric that captures the usefulness of a representation for the downstream prediction task
- The GULP metric measures the discrepancy between a pair of representations obtained via ridge regression estimators.
- GULP is invariant to orthogonal transformations, and is zero between two orthogonal representations


**Questions:**

Should you compare your method with probing methods in the NLP literature ? https://nlp.stanford.edu/~johnhew/interpreting-probes.html

**Limitations:**

Can’t find anything other than the ones mentioned by the authors.


**Strengths And Weaknesses:**

Strengths
- Proposing metrics for evaluating embedding is an important area for progressing AI
- Suitable theoretical analysis is provided for the properties of the metric
- Suitable related works are reviewed and compared
- Suitable experimental setups and results are provided

Weaknesses
- Various related works are reported in the introduction, however only 3 of these are compared in the experiments.

---

> ### Author Response · Authors · 2022-08-02
> **Response to Reviewer d1FQ**
>
> We thank the reviewer for their feedback, and respond to each point individually below.
>
> **Comparison to related work**
>
> We compare to Procrustes, CCA, CKA, and PWCCA, which matches the baselines tested in e.g. Ding et al. 2021.
>
> **Comparing method with probing in NLP**
>
> Thank you for this suggestion. Indeed, while we use probing ideas to derive distances between representation distances, it is natural to ask whether we can go the other way around, and in particular, examine the effect of ridge regularization on probing conclusions. This is an enticing direction for future research indeed. We will add a sentence to this effect in the conclusion of a camera-ready version.

---

### Official Review · Reviewer_QQnx · 2022-07-10

**Rating:** 6
**Confidence:** 4
**Soundness:** 3 good
**Presentation:** 3 good
**Contribution:** 3 good

**Summary:**

This paper proposes Uniform Linear Probing (GULP), a family of distance measures for the representations learned with neural networks, considering the representation performance. GULP satisfies the triangle inequality, shows a bounded sample complexity, and ignores the difference caused by orthogonal transformations.

GULP uses the idea of linear probing. Different from the works in literature, which require hand-selected prediction tasks, GULP naturally considers the prediction of the learned network as the target and measures the difference between predictions of two learned neural networks given the same dataset.


**Questions:**

Please see **Strengths And Weaknesses**. One concern is related to the question above it, so I put them together.

**Limitations:**

Yes,  the authors addressed the limitations and potential negative societal impact.

**Strengths And Weaknesses:**

The paper shows several **strengths**:

1. The method proposed is new and contributes to the distance measure between representations; even the representations are trained to have different numbers of dimensions.

2. There are theoretical results provided, suggesting the method is sound. The method shows a bounded sample complexity and stays sound when representations are different because one is an orthogonal transformation of the other, which is a common case in neural network training.

3. There is an analysis of the sensitivity of the newly introduced parameter ($\lambda$). Both empirical and theoretical evidence are provided.

4. The submission is well organized.

Though the topic studied by this paper is important and theoretical analysis has been provided to make the proposed method sound, I have the following **concerns and questions**.

1. In the introduction, the paper claims that GULP considers the representation's performance on downstream prediction tasks. However, in Definition 1, which defines the GULP distance, the equation uses two linear transformations, $\beta_{\lambda}$ and $\gamma_{\lambda}$. According to the equation above Definition 1, $\beta_{\lambda}=argmin_{\beta} \mathbb{E}\[ (\beta^T \phi(X) - Y)^2 \] + \lambda\| \beta \|^2$, $\beta$ is learned on the given dataset, which should come from the source task. Though the paper indicates the label $Y$ is assigned randomly in Section 4.1, the input space remains the same. I assumed $\gamma$ is learned the same way but could output a representation on a different dimensional space. Downstream tasks could be different from the source task, including the input distribution. Thus the performance could be very different. Is the assumption the source task and downstream tasks remain the same input space? Otherwise, it is not clear why the learned $\beta_{\lambda}$ and $\gamma_{\lambda}$ can be predictive of the performance on downstream tasks.

2. It is not clear to me how the centralization and normalization conditions are ensured during training in practice. The matrix $\beta$ and $\gamma$ are learned with an L2 constraint, which comes with a weight $\lambda$. With a more strict constraint, i.e., a larger $\lambda$, it seems like it would be harder to meet the request that the length of the learned representation is normalized to equal to 1. Assuming the ground truth $Y$ to be large and the norm of $\beta$ is restricted, the value of the learned representation will tend to be large. Or, is the condition of tuning $\lambda$ that finds the $\lambda$ that the learned representation is closest to meeting the centralization and normalization assumption?

3. There is also another concern related to the point above. Different $\lambda$ could lead to very different distance measures (Figure 2). Therefore an inappropriate $\lambda$ will affect the following evaluation of the distance between representations, thus affecting the empirical result of the proposed method.

---

> ### Author Response · Authors · 2022-08-02
> **Response to Reviewer QQnx**
>
> We thank the reviewer for their feedback, and respond to each point individually below.
>
> **Clarification on the source and downstream tasks (Point 1)]**
>
> First, we hope to clear up any potential confusion with a brief summary of GULP. As inputs, we are given two representations, $\phi$ and $\psi$. From the perspective of our method, it does not matter how $\phi$ and $\psi$ were obtained: part of our contribution is to think of a representation as a function, without an associated source task. This often matches practice: many users download generic, pretrained representations (such as CLIP representations) to use for their own downstream tasks. When we define the GULP pseudometric between representations, this is done with respect to a target (downstream) data distribution P_X over the inputs $X$. This is the data distribution on which one would like to use either representation for transfer learning. In other words, when we define the distance between representations $\phi$ and $\psi$, we implicitly mean the distance between representations ($\phi$, $P_X$) and ($\psi$, $P_X$). For notational convenience, we did not explicitly write this dependence on $P_X$ in $d_\lambda(\phi,\psi)$.
>
> When computing GULP with respect to $P_X$, the labels $Y$ are not assigned randomly; this was simply a synthetic example in Section 4.1, and in response to Reviewer JZhG, we have added a more realistic instance of generalization; see Figure 11. In fact, the GULP distance does not assume any distribution for the labels $Y$; it provides a worst-case bound over all norm-bounded labeling functions.
>
> Returning to your terminology, there are possibly three datasets: (1) one on which a representation $\phi$ is learned (in our numerical experiments, we often do not need access to this dataset since we employ pretrained representations $\phi$, (2) one on which a representation $\psi$ is learned, and (3) a third dataset, used to perform a Monte Carlo approximation of the expectation. We believe you call (1) and (2) “source” datasets, and (3) the “target” or “downstream” dataset. The “target” dataset may have nothing to do with the “source” dataset used to train \phi, but it is inherently associated to the prediction task on which this representation is to be used.
>
> In conclusion, it seems that the answer to your question is “no”: the “source” and downstream tasks need not be the same in our setting. Downstream predictions are done using $\beta^T\phi(X)$ or $\gamma^T\psi(X)$, respectively, where $X$ is indeed the same in both cases. Of course, the practitioner’s choice to use $\phi$ and $\psi$ will make more sense when the “source” and “target” datasets are related in some way (this is the principle behind transfer learning), and for simplicity, all of our experiments use the same “source” and “target” distributions $P_X$ (i.e. ImageNet, MNIST, or CIFAR), but this is not a requirement. Finally, the reviewer is correct that $\gamma$ is learned in the same way, and can output a representation on a different dimensional space. The ability to compare representations of different dimensionalities is a helpful feature of GULP.
>
> **Centering and normalizing (Point 2)**
>
> Please note that $\beta$ and $\gamma$ are vectors, not matrices. Moreover, they are not centered or normalized. Instead, it is the representations $\phi$ and $\psi$ which are centered and normalized. These representations are not learned; instead they are inputs to GULP’s distance computation. Approximate centering and normalizing of $\phi$ and $\psi$ can thus be performed easily given access to the downstream task dataset (or distribution, $P_X$, from the previous part’s response). This is a classical pre-processing step in two-input functions to make sure that the two inputs are commensurate.
>
> **Choice of regularization parameter (Point 3)**
>
> We view the regularization parameter as a perk of GULP, rather than a shortcoming. A practitioner is free to choose a larger $\lambda$ or a smaller $\lambda$, depending on the degree of specificity required by the application; as noted by Reviewer JZhG, this is not unlike the question of choosing a kernel for CKA. Tuning the parameter $\lambda$ is indeed an important question, and our numerical experiments indicate a moderate sensitivity to it. The choice of tuning parameters in ridge regression has been the object of numerous studies in statistics, and several heuristics such as cross-validation or the L-curve method are now classical. That being said, while running experiments, we find this tunability to be a useful feature for the purpose of visualization. Indeed, tuning the parameter lambda from $0$  to $\infty$ enables a multi-scale visualization of the space of representations, as illustrated in Figures 12, 13, and 21. We believe that this feature is key to discovering similarities that may arise at different scales, so one may not wish to fix a single regularization parameter $\lambda$.

---

> > ### Comment · Reviewer_QQnx · 2022-08-10
> > **Reply**
> >
> > I would like to thank the authors for the detailed response. I have increased the score above.

---

### Official Review · Reviewer_JZhG · 2022-07-12

**Rating:** 7
**Confidence:** 4
**Soundness:** 3 good
**Presentation:** 4 excellent
**Contribution:** 3 good

**Summary:**

The submitted paper proposes GULP, a family of dissimilarity measures between representations. GULP is based on the squared error between predictions of a ridge regression solution on the two representations. Each value of the ridge regression parameter $\lambda$ corresponds to a dissimilarity measure in this family. It is shown that the dissimilarity measures in this family satisfy properties of a pseudometric. Connections and differences to CKA and CCA are discussed in detail. Experiments study if similarity of representations in terms of GULP corresponds to generalization performance of linear regression and logistic regression on these representations, if similar architectures find similar representations, and if networks with the same architecture but different initializations find similar representations after training on the same task.

**Questions:**

See weaknesses section

**Limitations:**

See weaknesses section

**Strengths And Weaknesses:**

The provided metric has desirable properties of a dissimilarity measure of neural networks (invariance to rotation and isotropic scaling and capturing similarities in architecture) and the paper provides an unbiased estimator. The presentation is clear and related work is extensively discussed. I have not checked the proofs in the appendix but the math in the main paper is correct. I am voting for acceptance.

The following are my major comments:

1. One of the claims is that GULP captures generalization performance of linear regression. In Section 4.1, this claim is tested only on images with random labels. I find this one experiment unconvincing and I recommend either improving this experiment or rescinding the claim. Generalization and other aspects of the model's performance can be completely different if a real-world task is used instead of zero-mean Gaussian labels. Unfortunately there are not many benchmarks for regression compared to classification, which might be why the authors chose this synthetic task. There are some well-known datasets on age estimation that can be used in this section.

2. CKA provides the stated benefits of GULP other than satisfying pseudometric properties. A discussion on why it is important for a similarity measure of neural networks representations to satisfy these properties can help with motivating GULP. Are there currently any theoretical results or algorithms in this context that require these measures to be pseudometric?

3. GULP is a family of dissimilarity measures. How similar two representations are under GULP depends on how the hyperparameter $\lambda$ is tuned. This is not a critical problem to me. Similarity of representations is a loosely defined notion and different applications would call for different measures of similarity. Even CKA comes in different forms depending on its kernel (linear, RBF, etc.). For GULP, it is possible that a scalar value can be selected or constructed from this family on the specific application. I would suggest including more heuristic or theoretically grounded guidelines for choosing $\lambda$ in specific applications.

Minor comments:
- Bringing PROCRUSTES in the main text helps with understanding the results
- line 217: "with $\lambda=1$ resembles"
------------------------------------------------------
Update: The revision provides a realistic experiment about generalization with solid results and addresses my main concern. The discussion on the benefits of a pseudometric in the response is also helpful and convincing. I increased the score to 7.

---

> ### Author Response · Authors · 2022-08-02
> **Response to Reviewer JZhG**
>
> We thank the reviewer for their feedback, and respond to each point individually below.
>
> **Generalization experiment (Point 1)**
>
> We thank the reviewer for suggesting age estimation, and have rerun the experiment of Section 4.1 on the UTK face dataset. As shown in Figure 11 in the modified submission, GULP also predicts generalization better than baseline distances on this age regression task, with the optimal value for $\lambda$ corresponding to the regularization in the ridge regression itself.
>
> **Benefits of a pseudometric (Point 2)**
>
> GULP is a natural generalization of CKA-based similarity measures. As such, we do not advocate against CKA, but instead that GULP is a theoretically grounded alternative with direct interpretation  More to the point of similarity measures vs. pseudometrics, the reviewer is correct that there are some advantages of a pseudometric (though this discussion is perhaps beyond the scope of the present paper). For example, pseudometrics are amenable to visualization tools, like multidimensional scaling (MDS),and more generally to geometric tools such as near-isometric embeddings. In particular, many such geometrically-driven methods rely on the triangle inequality.  For example the geometry captured by a pseudometric is readily amenable to various algorithmic primitives such as  approximate nearest neighbors. Finally, the pseudometric approach better serves the goal of understanding the landscape of all representations --- to ultimately navigate it and optimize functionals over this space. Indeed the optimization toolbox largely relies on a classical metric structure.  In conclusion,  while both point-of-views, similarity and pseudometrics, have proved useful, we find the latter to be a more natural object to understand the space of representations from a geometric perspective.
>
> **Tuning the regularization parameter (Point 3)**
>
> Indeed, tuning the parameter $\lambda$ is an important question, and our numerical experiments indicate a moderate sensitivity to it. The choice of tuning parameters in ridge regression has been the object of numerous studies in statistics, and several heuristics such as cross-validation or the L-curve method are now classical. That being said, while running experiments, we find this tunability to be a useful feature for the purpose of visualization. Indeed, tuning the parameter lambda from $0$  to $\infty$ enables a multi-scale visualization of the space of representations, as illustrated in Figures 12, 13, and 21. We believe that this feature is key to discovering similarities that may arise at different scales, so one may not wish to fix a single regularization parameter $\lambda$.
>
> **Procrustes definition (Comment 1)**
>
> Thank you for bringing this to our attention. We will add the definition of Procrustes to the main text in a camera-ready version (using the extra page).

---

### Author Response · Authors · 2022-08-02
**Summary of Revisions**

We thank all reviewers for their thoughtful feedback. We have made a few additions to the submission in response, and summarize them here:
1. We replicate Figure 1 using the MNIST dataset (and the same ImageNet architectures for generating representations of MNIST), and observe that even on this simpler dataset, the architectures remain roughly clustered by type (Figure 18).

2. We replicate the original generalization experiment from Section 4.1 on the more realistic regression task of prediction age from faces on the UTKFace dataset, which can be found in Section B.4 of the Appendix. Once again, we find that GULP predicts generalization at least as well as competing measures.

3. Figure 13 shows how embeddings of CKA, Procrustes, and GULP distances spatially organize fully-connected ReLU networks trained on CIFAR with varying widths and depths (number of hidden layers). Similar to the experiments on MNIST with varying widths and depths (Figure 12), networks are organized linearly in terms of increasing depth and are closest to each other when layer widths become large.

4. Figure 14 reproduces the dendrogram from the original submission, which visualized the relationship between different ImageNet architectures, in a more readable format. We will move this figure to the main body in a camera-ready version.

5. Figure 21 of Section B.8 of the Appendix is a new experiment comparing the intermediate layer representations of BERT networks on the MultiNLI dataset, as a function of network depth. We find that the GULP distance arranges the hidden layers of all networks linearly in order from their input layer to their output layer. Tuning the parameter $\lambda$ allows us to make distinctions between earlier and later layers of the network architectures.

---

### Meta-Review · Area_Chair_QGMM · 2022-08-27

**Recommendation:** Accept
**Confidence:** Certain

**Metareview:**

All reviewers are in agreement that this is an interesting theoretical and empirical contribution and a useful tool in better understanding neural networks.

**Award:**

No

---

### Decision · Program_Chairs · 2022-09-14

Accept